# Prevalence of unmasked and improperly masked behavior in indoor public areas during the COVID-19 pandemic: Analysis of a stratified random sample from Louisville, Kentucky

Seyed M. Karimi[1,2]* , Sonali S. Salunkhe[1], Kelsey B. White[1], Bert B. Little[1], W. Paul McKinney[3], Riten Mitra[4], YuTing Chen[2], Emily R. Adkins[1], Julia A. Barclay[1], Emmanuel Ezekekwu[1], Caleb X. He[5], Dylan M. Hurst[6], Martha M. Popescu[7], Devin N. Swinney[1], David A. Johnson[1], Rebecca Hollenbach[2], Sarah S. Moyer[1,2], Natalie C. DuPré[8]

1 Department of Health Management and Systems Sciences, School of Public Health and Information Sciences, University of Louisville, Louisville, Kentucky, United States of America, 2 Louisville Metro Department of Public Health and Wellness, Louisville, Kentucky, United States of America, 3 Department of Health Promotion and Behavioral Sciences, School of Public Health and Information Sciences, University of Louisville, Louisville, Kentucky, United States of America, 4 Department of Bioinformatics and Biostatistics, School of Public Health and Information Sciences, University of Louisville, Louisville, Kentucky, United States of America, 5 Department of Political Science, College of Arts and Sciences, University of Louisville, Louisville, Kentucky, United States of America, 6 Department of Psychological and Brain Sciences, College of Arts and Sciences, University of Louisville, Louisville, Kentucky, United States of America, 7 Department of Anthropology, College of Arts and Sciences, University of Louisville, Louisville, Kentucky, United States of America, 8 Department of Epidemiology and Population Health, School of Public Health and Information Sciences, University of Louisville, Louisville, Kentucky, United States of America

☯ These authors contributed equally to this work.
* Seyed.Karimi@louisville.edu

**Data Availability Statement:** We have uploaded a de-identified dataset that can be used to replicate

## Abstract

Wearing a facial mask can limit COVID-19 transmission. Measurements of communities' mask use behavior have mostly relied on self-report. This study's objective was to devise a method to measure the prevalence of improper mask use and no mask use in indoor public areas without relying on self-report. A stratified random sample of retail trade stores (public areas) in Louisville, Kentucky, USA, was selected and targeted for observation by trained surveyors during December 14–20, 2020. The stratification allowed for investigating mask use behavior by city district, retail trade group, and public area size. The total number of visited public areas was 382 where mask use behavior of 2,080 visitors and 1,510 staff were observed. The average prevalence of mask use among observed visitors was 96%, while the average prevalence of proper use was 86%. In 48% of the public areas, at least one improperly masked visitor was observed and in 17% at least one unmasked visitor was observed. The average prevalence of proper mask use among staff was 87%, similar to the average among visitors. However, the percentage of public areas where at least one improperly masked staff was observed was 33. Significant disparities in mask use and its proper use were observed among both visitors and staff by public area size, retail trade

this study's results. The dataset is a csv file named "Louisville Mask Survey Data - Dec 2020" and is included among the Supporting information.

**Funding:** This study was funded by the Louisville Metro Department of Public Health & Wellness through the Coronavirus Aid, Relief, and Economic Security Act (the CARES Act). The funder did not play any role in the study design, data collection and analysis, decision to publish, and preparation of the manuscript. The authors who received the award were Sonali S. Salunkhe, Kelsey B. White, Emmanuel Ezekekwu, Emily R. Adkins, Julia A. Barclay, Caleb X. He, Dylan M. Hurst, Martha M. Popescu, and Devin N. Swinney.

**Competing interests:** No authors have competing interests.

type, and geographical area. Observing unmasked and improperly masked visitors was more common in small (less than 1500 square feet) public areas than larger ones, specifically in food and grocery stores as compared to other retail stores. Also, the majority of the observed unmasked persons were male and middle-aged.

## Introduction

Transmission of respiratory viral infections like COVID-19 can be reduced considerably by using a facial mask, especially in indoor public areas [1–8]. Therefore, as COVID-19 became a widespread pandemic in 2020, countries, states, and local municipalities began requiring facial coverings. Many governors within the United States implemented executive orders requiring masks within public indoor areas throughout 2020 [9]. The state of Kentucky mandated public masking on July 9th, 2020, and in March 2021 renewed the executive order until at least July 2021 [10, 11].

The requirement for facial coverings potentially limits the spread of the virus by asymptomatic and pre-symptomatic individuals who may cause nearly 60% of COVID-19 cases [12, 13]. While the importance of mask-wearing is emphasized, understanding mask-wearing behavior and practices in the community can inform public health policies. However, common methods of measuring mask-wearing yield significant inaccuracies, particularly surveys based on self-report [14–18] as they result in attenuation bias. When non-interventional observations of mask use replace self-reporting, non-representative sub-populations (e.g., university students and clinic population) have been studied [19–21]. Observational studies that do not focus on a specific sub-population or improper masking are scarce [22]. A study conducted in Wisconsin, USA, in summer 2020 approximated a 6% improper mask-wearing rate before the state's store mask mandate and suggested it decreased to 3% with a store mandate [23].

This study's primary aim was to develop a method to measure mask-wearing behavior in public areas accurately. To this purpose, visitors and staff of a representative sample of indoor public areas (PAs) in the city of Louisville (estimated 2019 population: 766,757) [24], Kentucky, USA, were observed. The representativeness of PAs was ensured by a stratified random sampling method, which allowed for assessing disparities in mask usage across city districts, industries, and PA sizes.

Also, in contrast to most existing mask usage studies [14–22], this study measured the prevalence of proper facial mask use in addition to the prevalence of facial mask use, as the effectiveness of mask mandates will be limited if a mask is not used properly. This study does not measure the prevalence of mask use and proper use before and after Kentucky's statewide mask mandate on July 9, 2020. Instead, it sought to identify the prevalence rate in December 2020 at the peak of the pandemic in the United States.

## Materials and methods

An observational survey was developed to assess facial mask-wearing behaviors in Louisville, Kentucky, during December 14–20, 2020. The observational aspect of this study meant that it relied on systematic observation of subjects' behavior in indoor public areas without intervention [25]. The observed PAs were selected using a stratified random sampling technique from the pool of retail trade businesses in the city. Surveyors were assigned subsets of the randomly selected PAs. They were trained to log their PA observation time in a standardized assignment sheet and to record observations with a standard online questionnaire (see "The questionnaire

and data collection method" and "The survey implementation" sections below for details). The filled questionnaires were cross-checked with surveyors' assignment sheets then the data was downloaded and refined to calculate four prevalence proportions of mask use: (1) the proportion of unmasked among visitors, (2) the proportion of unmasked among staff, (3) the proportion of incorrectly masked among visitors, and (4) the proportion of incorrectly masked among staff.

## Ethics committee approval

This study was reviewed on and determined by the University of Louisville Institutional Review Board that the study is exempt according to 45 CFR 46.101(b). This study was also approved through 45 CFR 46.116 (D), which means that it has been granted a waiver of informed consent. This study's approved IRB number is 20.0966.

## Definitions

In this study, an indoor public area was defined as an establishment (e.g., a business, store, or facility) where individuals can visit without an appointment or personal staff assistance. A PA's staff were identified by their uniform or clothing that displayed the PA's proprietary brand. If the PA's staff did not use uniforms, location and action (e.g., working behind counters, service area desks, and checkout registers) were used to identify them. A mask or facial mask included any type of facial covering—such as bandanna, cloth mask, neck gaiter, disposable surgical mask, cone-style mask, N95, and other respirators that may protect against aerosol transmission of infectious particles. A "masked" person wore a facial mask that covered both nose and mouth, an "improperly masked" person left either nose or mouth uncovered, and an "unmasked" person had neither nose nor mouth covered.

## Public area stratification criteria

Indoor PAs were selected using a random sampling technique stratified by the city district, industry, and PA size in order to observe mask-wearing in various parts of the community and in different types and sizes of PAs. Seven city districts—namely, (1) South & South West, (2) West Center, (3) North West, (4) North Center, (5) Central, (6) South East, and (7) East & North East—were constructed based on the geographical proximity of zip codes, demographic and median income (S1 Table and S1 Fig in S1 File). Notably, people who are Black are more highly concentrated in specific areas of the city, as the city suffers from a legacy of racial segregation highlighted discriminative zoning practices [26–28]. According to data from the 2010 census, the non-Hispanic Black population concentration was the highest in the North West district of the city (75%: 54%–93% in different zip codes). This district had the highest proportion of children (28% in 2010), and the median household income was the lowest ($22,848: $16,686–$27,565 in 2018) among all districts. The districts with the second and third largest share of non-Hispanic Black population were West Center (33%) and Central (23%). The West Center and Central districts' median household income was the city's second and third lowest: $37,469 and $43,911 in 2018, respectively. The non-Hispanic Black population in other districts was between 6% and 11% in 2010. The city's wealthiest district was East & North East, with a $91,141 median household income (S2 Table in S1 File). The U.S. median household income in 2018 was $63,179 [29].

The sampling strategy then considered industry for the second stratification criteria. Among Standard Industrial Classifications (SIC), Retail Trade (SIC division G, codes 52xxxx–59xxxx) was considered in this study [30]. Observing the other nine industrial classifications (e.g., Mining, Construction, Manufacturing, etc.) would have required business-specific

arrangements and resulted in higher research costs than what was available to this study. Among the subcategories of Retail Trade, Automotive Dealers and Gasoline Service Stations (SIC codes 55xxxx) and Eating and Drinking Places (SIC codes 58xxxx) were excluded to preserve the observational and non-interventional nature of the study, as most businesses in these two types of retail provide personal assistance to customers. Therefore, the study focused on the remaining subcategories of Retail Trade: Building Materials, Hardware, Garden Supply & Mobile Home Dealers (SIC codes 52xxxx), General Merchandise Stores (SIC codes 53xxxx), Food Stores (SIC codes 54xxxx), Apparel and Accessory Stores (SIC codes 56xxxx), Home Furniture, Furnishings and Equipment Stores (SIC codes 57xxxx), Miscellaneous Retail (SIC codes 59xxxx). The last category included drug stores and proprietary stores, liquor stores, used merchandise stores, and books stores, among others. A surveyor could conveniently visit a typical business classified under any of the six retail industries as a customer with no purchase and time cost for the surveyor.

The third stratification criterion was PA size so that the team could investigate if mask-wearing behavior differed in larger PAs where visitors may spend more time and cluster in greater numbers than smaller PAs.

## Businesses' data and surveying clusters

Data on information regarding the indoor PAs in the city were obtained from Data Axle in ArcGIS format [31]. A total of 4,648 PAs in Louisville were classified as Retail Trade (excluding SIC codes 55xxxx and 59xxxx) and were considered for observation. Information on the PAs was imported into the statistical software STATA 16.0 (STATACorp, LLC, College Station, TX, USA), which included zip code, SIC code, company name, primary address of the retail store, secondary address of the retail store, the retail location's latitude and longitude, and the square footage area of the retail store in ten categories.

Three variables representing stratification criteria were constructed: district, retail groups, and PA size. The city zip codes were grouped as described above, and the seven districts were formed (S1 Table and S1 Fig in S1 File). Six retail trade classifications were regrouped into four based on the similarity of trade types. Specifically, SIC codes 52xxxx (Building Materials, Hardware, Garden Supply & Mobile Home Dealers) and 57xxxx (Home Furniture, Furnishings and Equipment Stores) were combined to form the furniture, equipment, and home improvement stores group, and SIC codes 53xxxx (General Merchandise Stores) and 56xxxx (Apparel and Accessory Stores) were combined to form general merchandise and apparel stores group. 827 PAs were grouped into furniture, equipment, and home improvement stores, 996 in general merchandise and apparel stores, 878 in food and grocery stores, and 1,947 PAs in miscellaneous retail stores.

In terms of size, PAs were categorized into four size groups: PAs with square footage (1) less than 1499, (2) between 1500 and 2499, (3) between 2500 and 4999, and (4) greater than 5000. The specific recategorization was to increase the uniformity of PAs' distribution over size, given the footage brackets in the Data Axle dataset. In practice, PA size groups 1, 2, 3 and 4 included 1186, 1360, 954, and 1148 PAs, respectively.

Ultimately, 112 surveying clusters were formed. Each cluster included PAs specified by a combination of either of the seven districts, four retail groups, and four PA sizes (S3 Table in S1 File).

## Sample selection

The number of PAs in the 112 clusters ranged from 9 to 151. Tertiles of the distribution of the number of PAs in clusters were $\leq 25$, $> 25$–$44$, and $> 44$. Three PAs were randomly selected

without replacement from any cluster that fell in the first tertile, four PAs from the second tertile clusters, and five PAs from the third tertile clusters (S4 Table in S1 File). As a result, 447 PAs we randomly selected to be observed. Observing 447 PAs was perceived feasible based on the survey's cost assessment and funding. Using the same method, an extra 447 PAs were selected as a back-up in case any of the indoor PAs were non-operational.

To demonstrate the selected PAs' representativeness for observation, the proportion of each district in all selected PAs with the district's human and PA populations proportions (S5 Table in S1 File). The differences in proportions between the selected PAs and human population were small, between -2.7% and 2.7% across the seven districts, and the differences between selected PAs and PA population proportions were also small, between -3.8% and 4.3%. This suggested that the selected PAs were representative of the population distribution and PA distribution across districts.

As a more detailed representativeness test, the difference in each district's share in all selected PAs of a specific size was calculated with the district's human and PA populations shares. Therefore, two measures of error (equal to the absolute value of the two differences) were calculated for each of the 28 district-PA size clusters. When the shares of selected PAs in the districts were compared to the districts' human population shares, the mean error was 1.98%, and quartiles of the errors were 1.43%, 1.95%, and 3.05%, respectively. Comparing the share of selected PAs in the districts to the districts' PA population shares resulted in a mean error of 2.98%, with quartiles cut-points of 1.24%, 2.96%, and 4.48%.

## Surveyors' backgrounds and training

This study employed ten trained surveyors. Five of the surveyors were undergraduate students, 4 were public health doctoral students, and 1 faculty member from the university conducting the study. Among the group, 4 were men and 6 were women. All participated in a 1 hour orientation training prior to observation to ensure uniform collection efforts. Training included a review of the definitions of a mask, an unmasked person, and an incorrectly masked person and instructions for personal protective equipment use, conducting observations, and instructions on data entry. All received surgical masks, hand sanitizer, and face shields to protect them during observation. None of the surveyors were vaccinated prior to observation. Surveyors received monetary compensation for hours observing and mileage accrued during observations.

## Surveyors' assignment sheet

Each surveyor observed mask use in a subset of the randomly selected 447 PAs and a subset of the 447 back-up PAs that were also randomly selected. A surveyor's assignment sheet was a Word document that included two tables: a table of selected PAs and a table of reserve PAs. Each table had six columns: Unique ID, PA Zip Code, Retail Group, PA Size, PA Name, PA Address, Observation Number, and Notes.

A surveyor was asked to observe only the table of selected PAs. If a selected PA was not operational, then the surveyor was asked to replace it with a PA in the same retail group and the same size from the reserve table.

The Unique ID column was pre-populated with three-digit numbers. Unique IDs were generated to connect the collected data from a PA to its other information reported in the Data Axle master dataset. Surveyors would insert the PA's unique ID into the online questionnaire; hence the unique ID will appear as a survey data column.

An observation number is another randomly generated number that a surveyor transferred from the online questionnaire to the assignment sheet. The observation number was used as

an extra identification tool to connect a PA questionnaire (hence the survey data) to the PA's notes on the assignment sheet. Notes on the assignment sheet may contain data correction remarks if necessary (e.g., if a surveyor inserted a piece of information about the PA incorrectly or a selected PA was not operational).

## The questionnaire and data collection method

The survey's questionnaire included three sections: (1) PA Information, (2) Observing Visitors, (3) Observing Staff, and each section included five questions. The first section included pre-answered questions on PA identification information, Date of Survey, and Time of Survey. Also, a surveyor was asked to enter the PA Name and PA Unique ID from the assignment sheet (S6 Table in S1 File).

In the second section, surveyors were asked to observe the mask use of up to ten visitors in the PA. If there were more than ten visitors, they needed to observe only ten of them, but if there were fewer than ten visitors, they needed to observe all of them. To avoid selection bias in the observation of visitors, the surveyors started the observation as soon as they entered the PA. If the observation was not possible right after entering the PA, they started the observation at a random time and location in the PA. In any case, they (1) did not start observation after seeing an unmasked or incorrectly masked visitor and (2) observed the first ten visitors in the view. The second and third questions asked the numbers of the unmasked and incorrectly masked among the observed visitors, respectively. The fourth and fifth questions asked the perceived sex and age-ranges (0–18, 19–24, 25–44, 45–64, and 65+ years) of the unmasked and incorrectly masked. In the third section of the questionnaire, surveyors were asked about mask use among the PA staff. The section was structured similarly to the second section (S6 Table in S1 File).

The reason for observing a specific number of visitors or staff was to establish a denominator for the calculation of unmasked and incorrectly masked prevalence rates. The authors' experience from a previous pilot observational survey of mask use showed that counting all visitors and observing their mask use behavior in large public areas were not practical [32]. Ten was selected as the maximum number of observed visitors or staff in a PA for two reasons. First, the practice surveys, conducted by three authors, showed that working with a multiple of 10 (e.g., 10 or 20) was more straightforward for a surveyor and less prone to error. Second, to select a multiple of 10, practice surveys that targeted the observation of 10 and 20 visitors or staff (maximum totals of 20 and 40 visitors and staff, respectively) were conducted. The results showed that keeping track of 40 (versus 20) people and recalling several pieces of information (total count, the number of unmasked, the number of incorrectly masked, and perceived sex and age of the unmasked and incorrectly masked) was challenging.

The survey's questionnaire was designed in Qualtrics (Qualtrics, Provo, UT), and its link was emailed to surveyors to record observations from a PA. When a questionnaire was submitted, the link could be used to start filling another questionnaire for another PA. Answers were stored and were transferred to STATA for statistical analyses.

The survey was conducted from December 14 through December 20, 2020. A surveyor visited each indoor PA as a customer between 10:00 AM to 6:00 PM EST and observed mask-wearing behavior. Surveyors spent 5 to 15 minutes in a PA, depending on the store size, and completed the electronic survey.

## Results and discussion

### Sample characteristics

During the survey week, 382 PAs were visited where the mask-wearing behavior of 3,590 persons (2,080 visitors and 1,510 staff) were observed. The largest number of PAs were observed

in the North Center district (n = 71, 19%) (S2 Fig in S1 File). This district also included the largest share of the city's population (20%) (S4 Table in S1 File). The smallest number of PAs were observed in the North West district (n = 32, 8%) (S2 Fig in S1 File); the district included 11% of the city's population (S4 Table in S1 File). The observed PAs included 19% furniture, equipment, and home improvement stores, 24% general merchandise and apparel stores, 35% food and grocery stores, and 23% miscellaneous retail stores (S3 Fig in S1 File). A majority of the surveyed locations were over 5000 square feet in size (43%) or between 2500 and 5000 square feet (21%) (S4 Fig in S1 File).

The distribution of the number of observed PAs across the study's clusters was not precisely the same as the distribution of the number of selected PAs (S4 and S7 Tables in S1 File). In effect, the difference between a district's share in total observed PAs and the district's share in total human and targeted PA populations was greater than that for the district's share in total selected PAs (S8 Table in S1 File). Specifically, the difference between the observed PAs and human population shares was between -4.2% and 5.6% across the districts, and the difference between observed PAs and PA population shares was between -4.9% and 5.6%.

The larger error in observation than selection was the result of two implementation challenges. Firstly, several smaller-sized PAs either were not operational or could not be observed without an appointment. Secondly, the number of PAs in furniture, equipment, and home improvement stores and in miscellaneous retail stores that could not be observed without an appointment was greater than such PAs in other trade groups.

The distributions of observations were fairly even throughout days of the survey week and hours of the days (S9 and S10 Tables in S1 File). Except for Friday (December 18, 2020) in which 6% of the 382 PAs were observed, the share of other days of the week in the total observed PAs were between 12% and 18%. About 34% of the PAs were observed on a weekend day, 66% on a weekday. Except for the earliest and the latest observation hours (10:00 AM to 11:00 AM and 5:00 PM to 6:30 PM, respectively) in which 6% and 9% of the observations were made, the share of other hours in total observations was between 12% and 16%.

### Mask wearing for public areas' visitors

The mean proportion of unmasked observed visitors across all of the observed PAs was 4% (Standard Deviation (SD) = 14%), and in 83% of the PAs, there were no unmasked visitors. Both mean and variation of incorrect mask usage were greater than those of no mask use: 14% wore a mask incorrectly (SD = 22%). In 52% of the PAs, there were no observations of incorrectly masked visitors (Fig 1a). Incorrectly masked and unmasked visitors were most frequently observed within small public areas, those consisting of less than 1,500 square feet, where on average 10% (SD = 27%) were unmasked and 17% were incorrectly masked (SD = 29%) (Fig 1a).

Unmasked visitors were less frequently observed in the visited PAs from furniture, equipment, and home improvement stores, where the unmasked prevalence among visitors was zero in 92% of them, and the mean prevalence was 2% (SD = 13%). In PAs from the other retail groups, the prevalence of no mask-wearing was about 4% to 5% (Fig 2a). The incorrectly masked visitors were most commonly observed in food and grocery stores than the other three observed retail groups. In 57% of food and grocery stores, at least one incorrectly masked visitor was among the observed visitors, and the mean proportion of incorrectly masked visitors was 18% (SD = 23%) (Fig 2a).

Unmasked visitors were observed most frequently in the South & South West district, where the mean prevalence of unmasked visitors was 8% (SD = 21%) (Fig 3a). Nonetheless, the prevalence of unmasked visitors was zero in 76% of the observed PAs in this district.

(a) Visitors

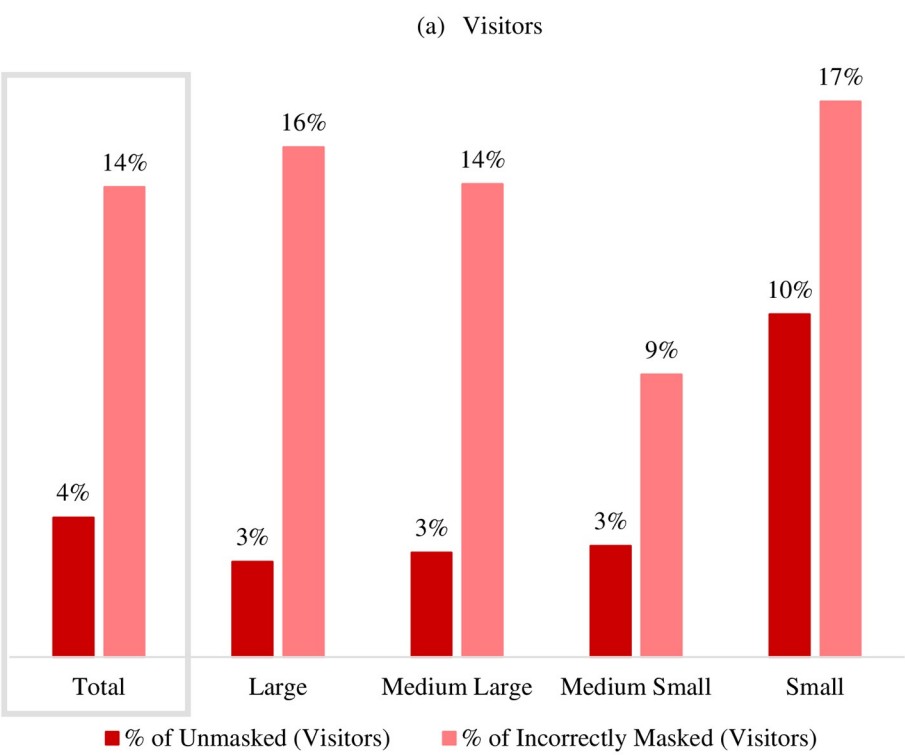

(b) Staff

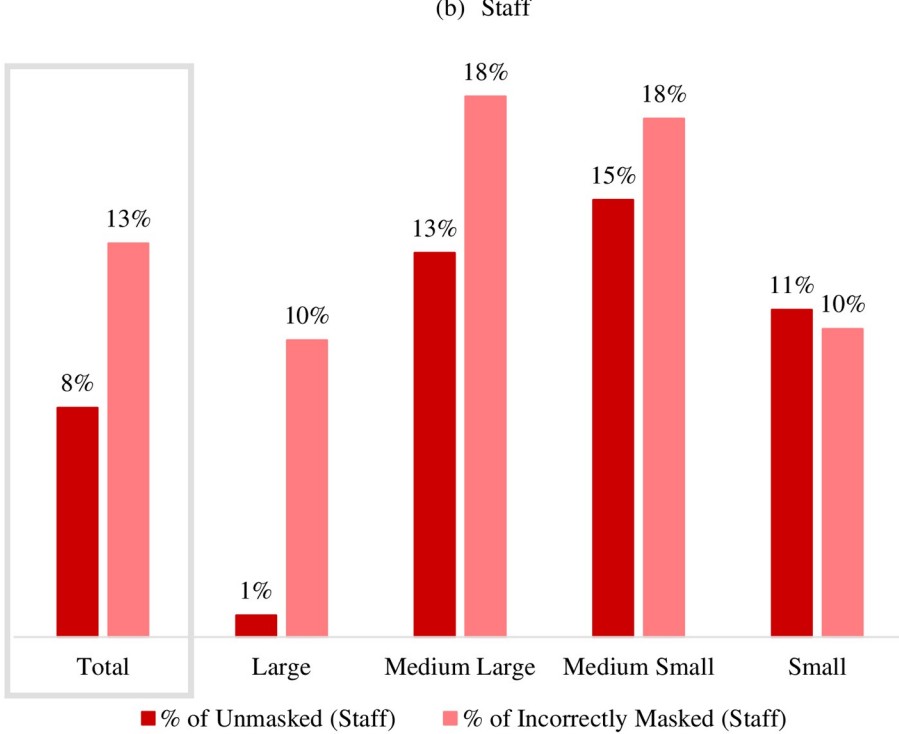

**Fig 1. The average prevalence of the unmasked and improperly masked among visitors and staff of Louisville indoor public by public area size, Dec. 14–20, 2020.** Large PAs are those with a square footage of at least 5000, between 2500 and 4999 sq ft for medium large, 1500 to 2499 sq ft for medium small, and 1499 or less for small.

(a)  Visitors

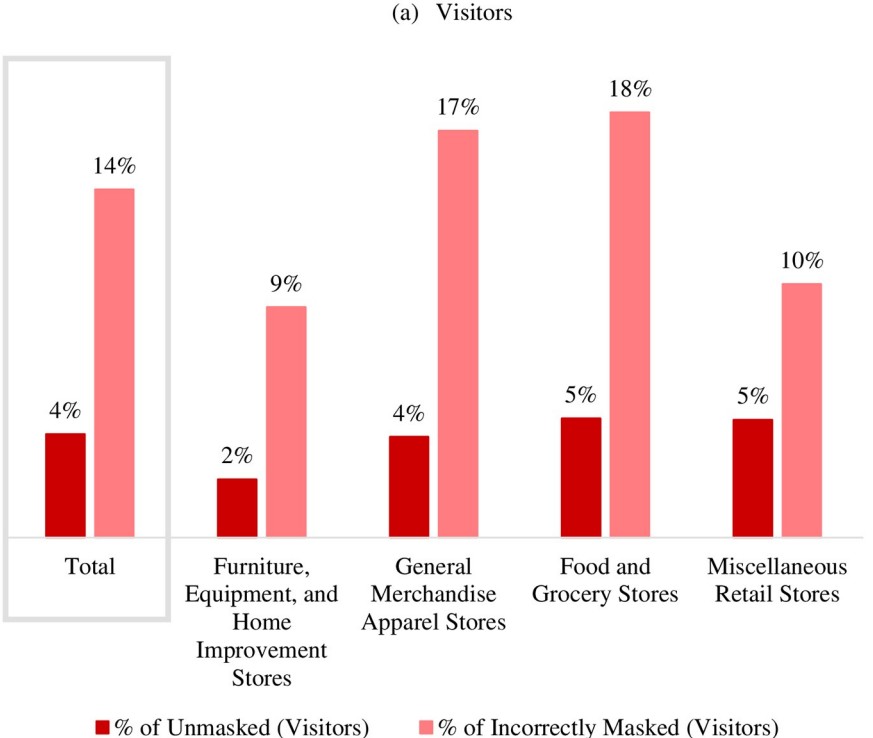

■ % of Unmasked (Visitors)    ■ % of Incorrectly Masked (Visitors)

(b)  Staff

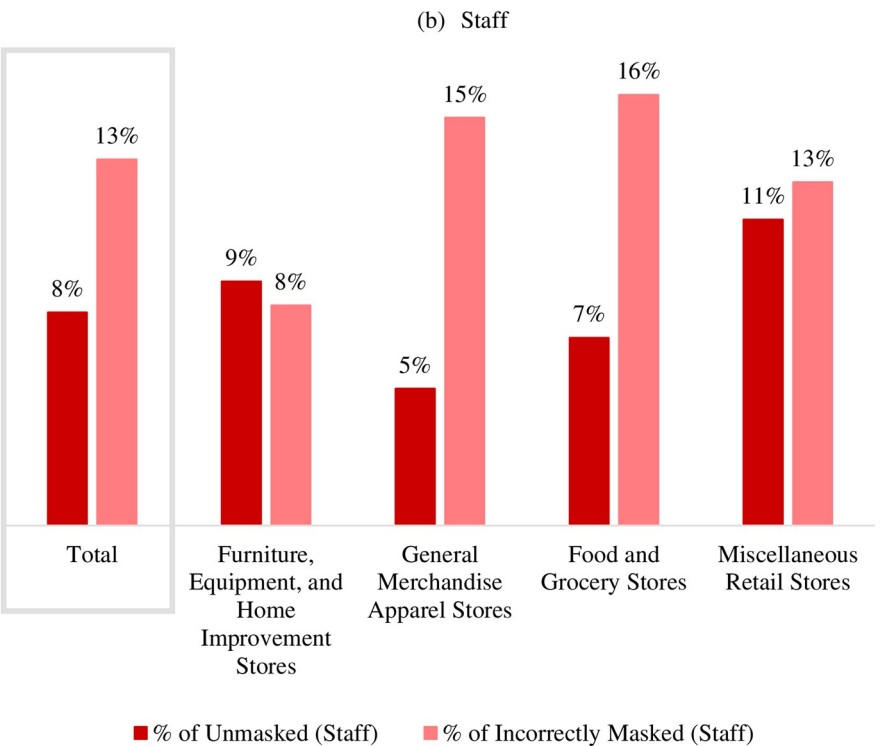

■ % of Unmasked (Staff)    ■ % of Incorrectly Masked (Staff)

**Fig 2. The average prevalence of the unmasked and improperly masked among visitors and staff of Louisville indoor public areas by retail trade group, Dec. 14–20, 2020.**

(a) Visitors

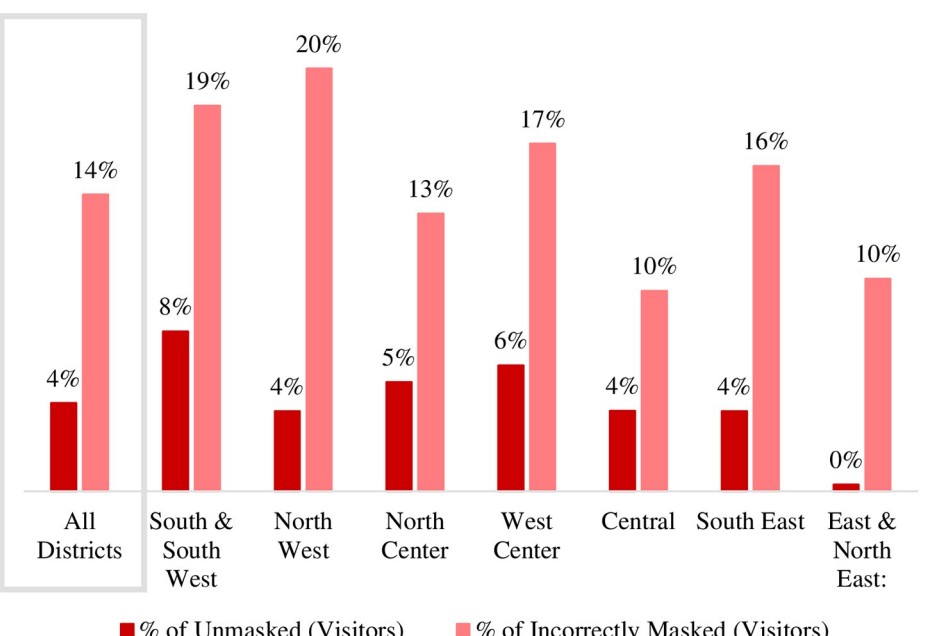

% of Unmasked (Visitors)    % of Incorrectly Masked (Visitors)

(b) Staff

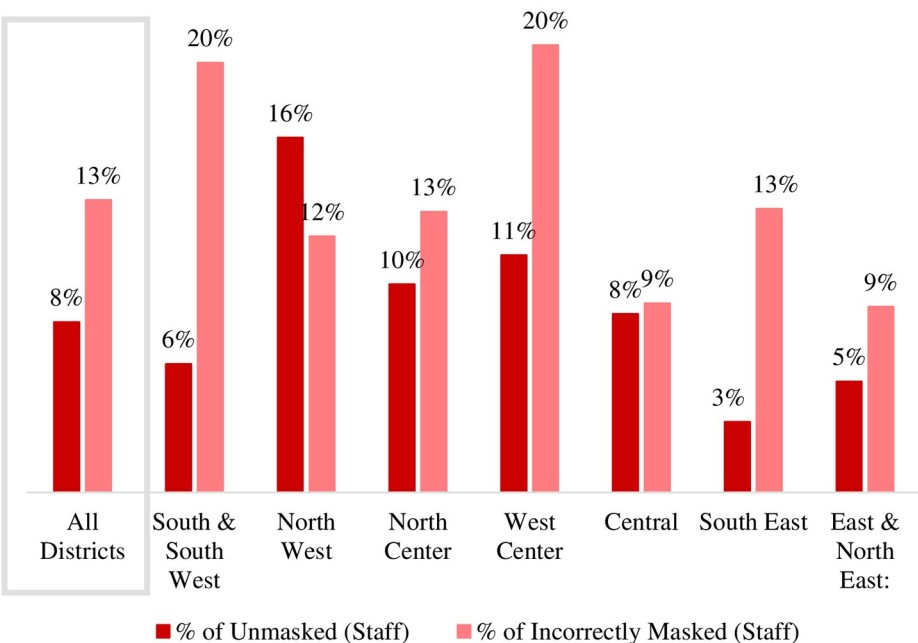

% of Unmasked (Staff)    % of Incorrectly Masked (Staff)

**Fig 3. The average prevalence of the unmasked and improperly masked among visitors and staff of Louisville indoor public areas by district, Dec. 14–20, 2020.**

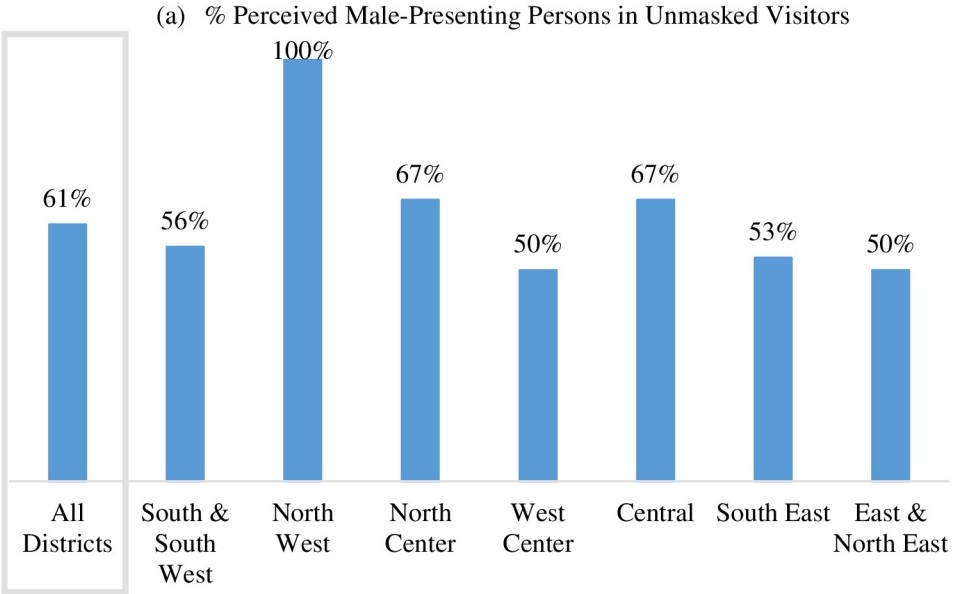

(a)  % Perceived Male-Presenting Persons in Unmasked Visitors

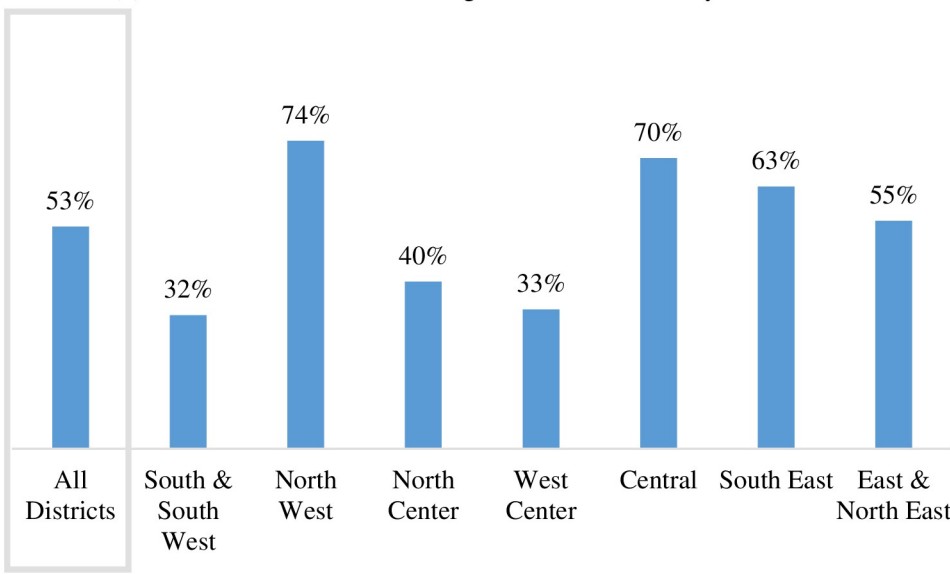

(b)  % Perceived Male-Presenting Persons in Incorrectly Masked Visitors

**Fig 4. Percentage of perceived male-presenting persons among unmasked and incorrectly masked visitors by district.**

Incorrectly masked visitors were observed most commonly in the North West district, where the mean prevalence of incorrectly masked visitors was 20% (SD = 29%), and at least one incorrectly masked visitor was observed in 57% of this district's PAs (Fig 3a).

Of all observed unmasked visitors, 61% were perceived male-presenting (Fig 4a). Among visitors wearing masks incorrectly, 53% were perceived male-presenting (Fig 4b). About 50% of the unmasked and 48% of the incorrectly masked visitors were perceived as middle-aged adults (Fig 5).

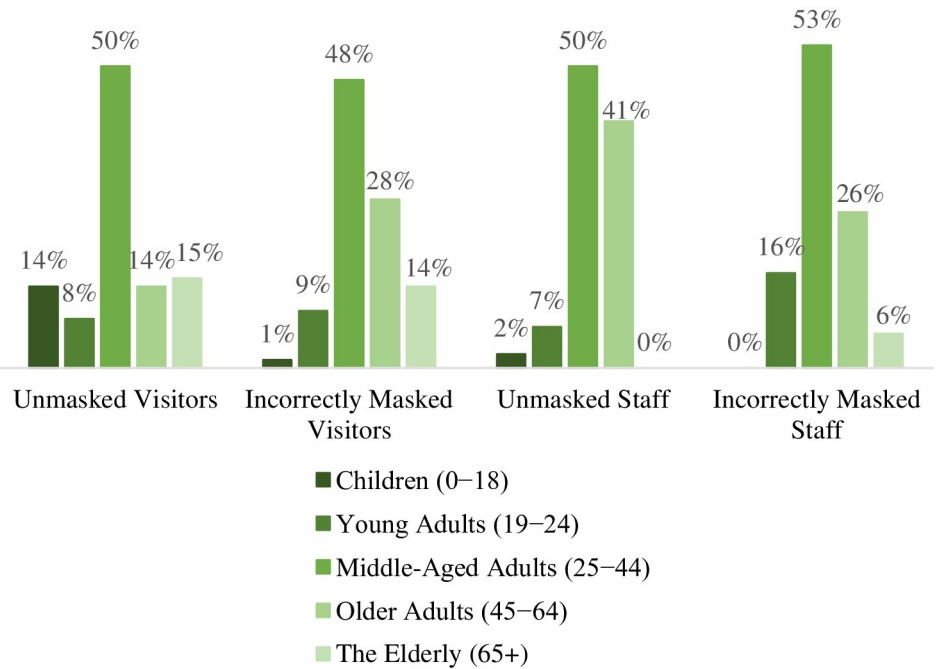

**Fig 5. The age distribution of the unmasked and incorrectly masked among visitors and staff.**

## Mask wearing for public areas' staff

Among the observed staff in the observed PAs, the mean prevalence of unmasked and incorrectly masked staff was 8% (SD = 25%) and 13% (SD = 25%), respectively (Fig 1b). The proportion of the unmasked and incorrectly masked were zero in 89% and 67% of the PAs, respectively.

Incorrectly masked and unmasked staff were most frequently observed within medium-small public areas, those with the square footage between 1500 to 2500, where the mean prevalence of no mask usage in staff was 15% (SD = 33%) and prevalence of incorrect mask usage in staff was 18% (SD = 30%) in medium-small PAs (Fig 1b).

Unmasked staff were less commonly observed in the visited PAs from general merchandise and apparel stores, where the unmasked prevalence among staff was zero in 93% of them, and the mean prevalence of unmasked staff was 5% (SD = 21%). In PAs from the other retail groups, the prevalence of no mask usage was higher, between 7% and 11% (Fig 2b). The incorrectly masked staff were most frequently observed in food and grocery stores than the other three observed retail groups. In 36% of food and grocery stores, at least one incorrectly masked staff was among the observed staff, and the mean prevalence of incorrectly masked staff was 16% (SD = 27%) (Fig 2b).

Unmasked staff were observed most commonly in the North West district, where the mean prevalence of unmasked staff was 16% (SD = 35%) (Fig 3b). Nonetheless, the prevalence of unmasked staff was zero in 68% of the observed PAs in this district. Incorrectly masked staff were observed most frequently in the South & South West and West Center districts—20% (SD = 33%) and 20% (SD = 31%), respectively (Fig 3b). Nonetheless, the proportion of incorrectly masked staff of these districts was zero in 65% and 59% of the observed PA in the two districts, respectively.

Of all observed unmasked staff, 50% were perceived male-presenting. Among observed incorrectly masked staff 42% were perceived male-presenting (S5 Fig in S1 File). About 50% of

the unmasked and 53% of the incorrectly masked visitors were perceived as middle-aged adults (Fig 5).

## Discussion

This study was conducted to estimate the prevalence of mask use and improper mask use in indoor public areas of the city of Louisville, Kentucky (USA) in December 2020 and to explore potential patterns in mask-wearing behavior. Results suggested there was a high prevalence of mask use: 96% of visitors and 92% of staff wore a mask in the 382 observed PAs. Improper masking appeared more common than not wearing a mask at all in December 2020. Among the masked visitors and staff, respectively, 86% and 87% used the mask properly, covering both the nose and mouth (Fig 1). The proportion of PAs where at least one unmasked staff was observed was smaller than the share of PAs where at least one unmasked visitor was observed (11% versus 17%). Similarly, the share of PAs where improperly masked staff were observed was smaller than the share of PAs where improperly masked visitors (33% versus 48%).

This research suggests that variation in the proper use of masks may exist depending on the size of the PA (Fig 1). Although previous research reports similar findings [32], why this occurs remains unknown. One explanation could be the way individuals follow perceived social norms [33]. Perhaps individuals more intentionally follow a norm in areas with more people; there is a greater expectation to follow a pre-set norm, such as wearing a mask correctly in larger rather than smaller public areas. Others suggest that health behaviors are more specifically connected to one's appraised threat such that the lower the appraised threat, the less likely the individual will adhere to protective behaviors [34, 35]. In other words, if one assesses a lower risk of contracting COVID-19 in a smaller establishment due to the location's smaller capacity, then they are less motivated to completely adhere to those protective behaviors [34]. Future studies examining mask-wearing behavior should include data points about individual behavior beliefs, protective beliefs, beliefs about infectious diseases, and misinformation beliefs [36].

The findings showed that unmasked visitors and incorrectly masked visitors ranged from 2% to 5% and 9% to 18%, respectively, across different retail stores (Fig 2). Also, unmasked and incorrectly masked staff in the retail stores ranged from 5% to 11% and 8% to 16%, respectively (Fig 2). A survey conducted by the Pew Research Center in August 2020 revealed that approximately 85% of people reported wearing a mask in July which increased from 65% in June [8]. One study reported that the odds of observing a person wearing a mask in an urban or suburban retail store were 4 times higher than in rural areas [23]. Despite mask mandates in place, some people either do not wear a facial mask or wear them incorrectly, thus putting not only others in their close proximity but also themselves at risk of contracting the infection due to transmission from asymptomatic COVID-19 carriers [23].

Significant geographical variation in the prevalence of mask use and improper use was observed in this study. The mean prevalence of unmasked persons varied from 0% to 8% in visitors and 3% to 16% in staff. Geographical variation in mask use was strongly correlated with income such that the districts with highest prevalence of unmasked and improperly masked individuals were among the most economically disadvantaged districts of the city. For example, the highest visitor improperly masked prevalence (20%) and the highest staff unmasked prevalence (16%) were recorded in the city's poorest district, the North West district, where the median family income was $22,848 in 2018. Conversely, the lowest visitor unmasked and improperly masked prevalence (0% and 10%, respectively) and the lowest staff unmasked prevalence (5%) were recorded in the richest district of the city, East & North East

where the median family income was $91,141 in 2018 (S2 Table in S1 File). Populations with low-income often face greater challenges obtaining resources to practice healthy behaviors and often face complex social contexts [37, 38]. The financial means to purchase or make an appropriately fitted mask may be an obstacle in lower income communities. Researchers have reported a higher prevalence of high-risk health behaviors among communities with low-income [39]. In other words, although smoking and physical inactivity differ from mask-wearing, the communities facing the greatest health challenges for systemic reasons are also struggling with masking. Further, researchers currently suggest that case and mortality rates are elevated for populations with complex health challenges, in areas health disparities already exist, and potentially for populations with low-income [40].

The results highlighted that a majority of the improperly masked or unmasked were middle-aged male-presenting adults in the approximate age range of 25 to 44 years (Figs 4 and 5). Finding a high prevalence of incorrect mask use among males coincides with previous research findings. A number of studies report that females are more likely than males to wear masks or wear masks appropriately [22, 23, 41]. One study suggests a connection between masculinity and mask-wearing behaviors [37] and other research suggests that the tendency to appropriately wear a mask is connected to one's caregiving responsibilities [42]. Masking-wearing as a public health practice may evolve as the utilization of seatbelts did beginning in the mid-1900s in the United States. If so, identifying the history of successful and unsuccessful public health practice campaigns will further public marketing for masking. For example, researchers have reported that young men are the most unlikely to follow seatbelt regulations [43–45], and others have suggested that seatbelt use is connected with one's perceived risk [46]. Even other public health prevention tools, such as condom use, has been connected with one's tendency to take risk or engage in impulsive behaviors [47]. This evidence suggests mask-wearing behaviors may face similar challenges.

The psychological factors that influence adherence to recommendations for public health behaviors often receive less attention than the medical factors. Although medical and public health practices have evolved throughout the pandemics of the past, human psychology has adapted less [48]. Attending to the psychological factors, such as those that lead individuals to engage in risky behaviors or send others into extreme isolation, warrant public health planning as well [48, 49]. Individuals' anxiety and beliefs about their own health influence adherence to hygiene practices and social distancing [48]. Future pandemic planning will need to consider planning for increased psychological needs and crises.

Political factors, cultural dynamics within a community, socioeconomic factors, media and social media, and governmental policies influence a community's adherence to health behavior recommendations. One survey identified the political and polarizing nature of mask-wearing among United States counties [50]. Louisville is an urban area within a predominately rural state with midwestern and southern history. Mask-wearing behaviors may differ between rural and urban settings as well [23]. Louisville voted for democratic representatives in the House of Representatives eight out of the past ten election cycles [51]. In 2018, Kentucky elected a democratic governor after 4 years with a republican governor [52] which speaks to the dynamic political and cultural context of the state. Thus, a wider sampling of Kentucky as a whole would provide a more representative snapshot of these behaviors and attend to the geographical and political variation in the state. Further, the evidence from international responses' also highlights how political dynamics and governmental policies impact a population's response to pandemic health recommendations [53]. Even more localized leadership, beyond a national or state level, can influence a community's response [53].

## Limitations

Only businesses classified as Retail Trade by SIC were observed in this study. Retail trade establishments account for 15% of all business establishments in the U.S. and about 14% in Louisville [30, 31]. Therefore, the results of this study cannot be generalized to mask-wearing behaviors in indoor public areas of non-retail trade businesses. Even among the retail trade, two major classifications—namely, (1) automotive dealers and gasoline service stations and (2) eating and drinking places, constituting about 5% of all businesses in Louisville—were excluded [31]. Observing eating and drinking places is especially important to understand the dynamics of the spread of respiratory infectious disease, as they are environments where masks are taken off, at least occasionally, and have been linked to an increase in COVID-19 cases [54].

The representativeness of the observed sample of PAs was not as complete as the study strategically planned. The median representativeness error (defined as the difference of the share of observed PA of a specific size from a district in total observed PAs of that size from the population share of the district) in the observed sample was 3.92%. Approximately half of the error could be attributed to the sample selection mechanism that resulted in a median representativeness error of 1.95%. The rest could be attributed to the implementation challenges. For example, some of the selected PAs were either non-operational or could not be visited by surveyors. This happened more often in small PAs (square feet less than 1499), especially in the West Center, North Center, and South East Districts.

The survey of mask use behavior was conducted at the height of the COVID-19 pandemic in the United States and the city of Louisville (S6 Fig in S1 File). If people's personal protection behavior is affected by the extent of the pandemic in the country and their community, one would expect that mask use behavior to be different at times with varying rates of infection and deaths from COVID-19. Therefore, the results of this study may not capture mask use behavior of citizens in other periods.

In addition, this study's results for sex and age-range of the unmasked and incorrectly masked need careful interpretation. For example, the majority of the unmasked and incorrectly masked visitors of the observed PAs were male-presenting and middle-aged adults. However, one cannot be certain that males and middle-aged adults exhibit the worst mask-wearing behavior without an interventional study design. In other words, the study introduced observer bias as a product of misjudging visitors' and staff's sex and age due to limited facial visibility. As a result, sex and age could have been misclassified to some degree. In addition, misclassification of mask use or correct use could have occurred if an observed visitor or staff member took off or moved face-covering temporarily during observation.

The type of facial mask (e.g., surgical, cloth, bandana, gators, or face shields) was not observed in the current study. In the pilot survey [55], the predominant masks used were surgical (50%) and cloth masks (50%) that did not differ across public areas or zip codes.

## Conclusions

The findings from this observational study showed that the incorrectly masked and unmasked visitors were frequently observed in small public areas (Square footage < 1,500). In contrast, the incorrectly masked and unmasked staff were more commonly observed in medium-small public areas (Square footage = 1500–2500). Both incorrectly masked visitors and staff were most regularly observed in food and grocery stores than other retail stores. Among the observed visitors and the staff, middle-aged adults made up the highest proportion of unmasked and incorrectly masked. Despite mask mandates in place, we observed a small proportion of visitors (4%) and staff (8%) that did not wear a facial mask or wore them incorrectly

(14% of visitors and 13% of staff). There is a continued need to improve awareness of the effectiveness of appropriate facial mask use, financial resources to provide masks, particularly in low-income areas, and education on correct ways to wear a mask.

## Supporting information

**S1 File.**
(DOCX)

**S1 Data.**
(CSV)

## Acknowledgments

We thank Craig H. Blakley, Christopher E. Johnson, Wanda D. Long, Eric Nunn, Darla D. Samuelsen, Christopher R. Tillquist, and Tammi A. Thomas for their critical helps with the project's logistics. We also thank the Commonwealth Institute of Kentucky (CIK) for providing access to Qualtrics to design this study's online questionnaire.

## Author Contributions

**Conceptualization:** Seyed M. Karimi, Sonali S. Salunkhe, Kelsey B. White, Bert B. Little, W. Paul McKinney, Riten Mitra, Natalie C. DuPré.

**Data curation:** Seyed M. Karimi, Sonali S. Salunkhe, Kelsey B. White, YuTing Chen, Emily R. Adkins, Julia A. Barclay, Emmanuel Ezekekwu, Caleb X. He, Dylan M. Hurst, Martha M. Popescu, Devin N. Swinney.

**Formal analysis:** Seyed M. Karimi, Sonali S. Salunkhe, Kelsey B. White, YuTing Chen, Natalie C. DuPré.

**Funding acquisition:** Seyed M. Karimi, Bert B. Little, Rebecca Hollenbach, Sarah S. Moyer.

**Investigation:** Seyed M. Karimi, Sonali S. Salunkhe, Kelsey B. White.

**Methodology:** Seyed M. Karimi, Sonali S. Salunkhe, Kelsey B. White, Bert B. Little, W. Paul McKinney, Riten Mitra, YuTing Chen, Natalie C. DuPré.

**Project administration:** Seyed M. Karimi, Sonali S. Salunkhe, Kelsey B. White.

**Resources:** Seyed M. Karimi, YuTing Chen, David A. Johnson, Rebecca Hollenbach, Sarah S. Moyer.

**Software:** Seyed M. Karimi, Sonali S. Salunkhe, Kelsey B. White, YuTing Chen.

**Supervision:** Seyed M. Karimi, Sonali S. Salunkhe, Kelsey B. White, Bert B. Little, W. Paul McKinney, Riten Mitra, David A. Johnson, Rebecca Hollenbach, Sarah S. Moyer, Natalie C. DuPré.

**Validation:** Seyed M. Karimi, Sonali S. Salunkhe, Kelsey B. White.

**Visualization:** Seyed M. Karimi, Sonali S. Salunkhe, Kelsey B. White, YuTing Chen.

**Writing – original draft:** Seyed M. Karimi, Sonali S. Salunkhe, Kelsey B. White.

**Writing – review & editing:** Seyed M. Karimi, Sonali S. Salunkhe, Kelsey B. White, Bert B. Little, W. Paul McKinney, Riten Mitra, Emmanuel Ezekekwu, Natalie C. DuPré.

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
