## [Decision Letter · Decision Letter 0]

19 Apr 2021

PONE-D-21-05854

Stratified Random Sampling Methodology for Observing Community Mask Use within Indoor Settings: Results from Louisville, Kentucky during the COVID-19 Pandemic

PLOS ONE

Dear Dr. Seyed M. Karimi,

Thank you for submitting your manuscript to PLOS ONE. After careful consideration, we feel that it has merit but does not fully meet PLOS ONE’s publication criteria as it currently stands. Therefore, we invite you to submit a revised version of the manuscript that addresses the points raised during the review process.

We look forward to receiving your revised manuscript.

Kind regards,

Cosme F. Buzzachera, Ph.D.

Academic Editor

PLOS ONE

Journal Requirements:

Please clarify whether the ethics committee waived the need for informed consent. If not, please explain why this was not necessary in this study.

We note that there appears to be a rendering error in Appendix Figure 3. Please revise this figure.

We note that data on the perceived sex of observed people were collected. Throughout the manuscript, please ensure that you consistently refer to observed males and females as "perceived female-presenting" and "perceived male-presenting" and please briefly discuss as a limitation the possibility that some of these perceptions may have been incorrect as no conversations were had with people being observed.

We noted in your submission details that a portion of your manuscript may have been presented or published elsewhere. “Some of the results reported in this manuscript were provided to the Louisville Metro Department of Public Health and Wellness (which funded the research) as an internal technical report. “

We note that you have indicated that data from this study are available upon request. PLOS only allows data to be available upon request if there are legal or ethical restrictions on sharing data publicly. For more information on unacceptable data access restrictions, please see http://journals.plos.org/plosone/s/data-availability#loc-unacceptable-data-access-restrictions.

6a) If there are ethical or legal restrictions on sharing a de-identified data set, please explain them in detail (e.g., data contain potentially sensitive information, data are owned by a third-party organization, etc.) and who has imposed them (e.g., an ethics committee). Please also provide contact information for a data access committee, ethics committee, or other institutional body to which data requests may be sent.

6b) If there are no restrictions, please upload the minimal anonymized data set necessary to replicate your study findings as either Supporting Information files or to a stable, public repository and provide us with the relevant URLs, DOIs, or accession numbers. For a list of acceptable repositories, please see http://journals.plos.org/plosone/s/data-availability#loc-recommended-repositories.

Thank you for stating the following in the Acknowledgments/Funding Section of your manuscript:

This study is funded by the Louisville Metro Department of Public Health & Wellness

through the Coronavirus Aid, Relief, and Economic Security Act (the CARES Act).

“YES

1. Sonali S. Salunkhe: Conceptualization, Methodology, Fieldwork organization, Surveyor, Data curation, Formal analysis, Supervision, Writing the original draft, Reviewing and editing

2. Kelsey White: Conceptualization, Methodology, Fieldwork organization, Surveyor, Data curation, Formal analysis, Supervision, Writing the original draft, Reviewing and editing

3. Emily R. Adkins: Surveyor

4. Julia A. Barclay: Surveyor

5. Emmanuel Ezekekwu: Surveyor

6. Caleb X. He: Surveyor

7. Dylan M. Hurst: Surveyor

8. Martha M. Popescu: Surveyor

9. Devin N Swinney: Surveyor”

We note that Appendix Fig 1 in your submission contain map images which may be copyrighted. All PLOS content is published under the Creative Commons Attribution License (CC BY 4.0), which means that the manuscript, images, and Supporting Information files will be freely available online, and any third party is permitted to access, download, copy, distribute, and use these materials in any way, even commercially, with proper attribution. For these reasons, we cannot publish previously copyrighted maps or satellite images created using proprietary data, such as Google software (Google Maps, Street View, and Earth). For more information, see our copyright guidelines: http://journals.plos.org/plosone/s/licenses-and-copyright.

1.            You may seek permission from the original copyright holder of Appendix Fig 1 to publish the content specifically under the CC BY 4.0 license. 

Please include your tables as part of your main manuscript and remove the individual files. Please note that supplementary tables (should remain/ be uploaded) as separate "supporting information" files

Please include captions for your Supporting Information files at the end of your manuscript, and update any in-text citations to match accordingly. Please see our Supporting Information guidelines for more information: http://journals.plos.org/plosone/s/supporting-information

Reviewers' comments:

Reviewer's Responses to Questions

**Comments to the Author**

1. Is the manuscript technically sound, and do the data support the conclusions?

Reviewer #1: Yes

Reviewer #2: Yes

Reviewer #3: Yes

2. Has the statistical analysis been performed appropriately and rigorously? 

Reviewer #1: Yes

Reviewer #2: Yes

Reviewer #3: Yes

3. Have the authors made all data underlying the findings in their manuscript fully available?

Reviewer #1: No

Reviewer #2: No

Reviewer #3: Yes

4. Is the manuscript presented in an intelligible fashion and written in standard English?

Reviewer #1: Yes

Reviewer #2: Yes

Reviewer #3: Yes

5. Review Comments to the Author

Reviewer #1: The manuscript is well written and brings up an important topic in the current context.

Yet, I have a some comments for consideration by the authors.

1) Are there any results in terms of mask type that could be presented in the results section?

2) Why only up to ten visitors were observed in each PA? And which criteria did the observer use to choose the ten visitors that he/she observed in PAs with more than ten visitors? In other words, how did the observer avoid any bias in selecting the ten visitors he/she observed? It is important to clarify it, as an observer could observe different proportions of unmasked and masked visitors depending simply on the procedures adopted.

3) Was the use of facial mask mandatory throughout the period of observation? I suggest providing this information at some point of the introduction and discussion.

4) What was the proportion of PAs that presented reinforcement signs? And did the existence of these reinforcement signs presented any influence on the use of facial mask?

Reviewer #2: The manuscript "Stratified Random Sampling Methodology for Observing Community Mask Use within Indoor Settings: Results from Louisville, Kentucky during the COVID-19 Pandemic” showed data about the use (prevalence of incorrect use) of masks during the COVID-19 pandemic period at the end of 2020. Previous studies have been suggested low levels of incorrectly used masks because of self-reported procedures. This study presented data that may represent better the “real world” scenario since it used an observational evaluation and not a self-reported procedure. I recommend accepting this manuscript after minor revision. Some comments and suggestions are described below:

## Comments on abstract and title

#1. Although the method is a great part of the manuscript, the information about the incorrect use of masks may be the most relevant news from the study. In this way, I suggest authors change the title for “Prevalence of Unmasked and Improperly Masked in Indoor Public Areas during COVID-19 Pandemic: Data Form Stratified Random Sampling Methodology For Observing Community”.

#2. The abstract is clear and presented the lack of information required in the field specifically solved in this manuscript (prevalence of incorrect use of masks is different from observational studies than self-declaration studies). In the abstract, it was informed that “The average mask use prevalence among observed visitors of the 382 visited public areas was 96%, while the average prevalence of proper use was 86%.”. It may be relevant to inform the total number of subjects observed in these 382 public areas.

## Comments on introduction/background

#3. The authors presented the background and the relevance of the study in the introduction adequately. The authors presented the arguments about the use of an observational procedure instead of a questionary to collected data from subjects about the correct use of masks. However, I recommend showing at least some results about the prevalence of incorrect use of masks from the studies cited in the introduction.

## Comments on methods

#4. For better replicability of this study in other cities and countries, I recommend the author inform the number and profile of the volunteers/observators. Were they students from university? Were they wearing special protection? Were they vaccinated before starting the field procedure? Some of these questions also imply the ethical aspects of the involvement of the volunteers.

#5. What was the period of the day (morning, afternoon…) of the observation. The profile of visitors may be different in the early morning in comparison with the end of the day.

#6. The authors informed that “The fourth and fifth questions asked the surveyor’s to perceive the sex and approximate age-ranges”. There was a previous training procedure to prepare the surveyors to identify the age of the subjects? Is there some limitations fro this identification because of the use of masks?

## Comments on results

#7. The results are clear and well presented. I suggest to authors to first present the “Mask wearing for public areas’ visitors” results (page 14) following by the “Mask wearing for public areas’ staff” (page 15) and after that to present the results described now on page 13.

## ## Comments on discussion

#8. Since the adhesion to health recommendation is influenced by many aspects, from political recommendations procedures, level of understanding of the population, media coverage, self-perception of risk, economic and educational status, I suggest to authors include some aspects in the discussion. It is clear the impact of the position of some governments, mainly in Brazil and USA, on the use of masks. I suggest including this scenario in the manuscript. Also, I suggest reading two references: Taylor S. The psychology of pandemics: Preparing for the next global outbreak of infectious disease. Newcastle upon Tyne: Cambridge Scholars Publishing. Cambridge Sch. 2019. This reference discusses many aspects that may interfere in the adhesion of health recommendations. Some parts of this discussion are also present in the discussion of this paper: Heck et al., Insufficient social distancing may contribute to COVID-19 outbreak: The case of Ijuí city in Brazil. PLoS One. 2021 Feb 17;16(2):e0246520. https://doi.org/10.1371/journal.pone.0246520

I hope that this revision helps the authors to improve the manuscript.

Sincerely,

Reviewer #3: Title: Stratified Random Sampling Methodology for Observing Community Mask Use within Indoor Settings: Results from Louisville, Kentucky during the COVID-19 Pandemic

Summary: The study under review explored a method for measuring the prevalence of mask-wearing and proper mask use in indoor public areas without relying on self-report. A stratified random sample of retail trade stores (public areas) in Louisville was selected and targeted for observation by trained surveyors during December 14−20, 2020. The stratification allowed for investigating mask use behaviour by city district, retail trade group, and public area size. The authors noted a high average mask use prevalence (96%) among observed visitors of the 382 visited public areas, with an average prevalence of proper use of 86%. The authors also noted a high average mask use prevalence (92%) among staff, with unmasked staff being observed in fewer public areas. Observing unmasked and incorrectly masked visitors were more common in smaller public places and food and grocery stores. The majority of the observed unmasked persons were male and middle age adults. The authors suggested a continued need to improve awareness of the effectiveness of appropriate facial mask use, financial resources to provide masks, particularly in low-income areas, and education on correct ways to wear a mask.

General comments: The authors are commended for a well-written manuscript. The arguments for the manuscript under review are timely and original. I believe there are minor concerns and issues with the manuscript in its current form that need to be addressed before being considered for publication. All my comments are included below. I hope you will find them to be constructive and helpful.

Minor Concerns: The primary concerns with the manuscript are presented below:

Introduction. The Introduction section is well-written but relatively short in general. The authors are encouraged to bring out the main points a little more firmly and insert necessary information. The authors are advised, for example, to include a priori research hypotheses with their respective references.

Materials and Methods. The study was conducted for a brief period of 2020 December. Could the study findings, therefore, be similar to other periods? Please comment. Additionally, how was the COVID-19-related scenario (n. of cases and n. of deaths) in the USA and Louisville area during the study? The study's results may, at least in part, depend on the pandemic scenario in the city/country. The authors are, therefore, suggested to take into account this inherent "limitation." The authors are also advised to insert a figure depicting the COVID-19 milestones, which could help the readers.

Is there any explanation/reason to observe 10 visitors in each PA? Please comment

Discussion. Face-covering misclassification could be an inherent limitation of this observational study. As mentioned by the authors, both sex and age could have been misclassified due to surveyor bias. Face-covering misclassification, however, could also occur if the mask was removed immediately before the observation. This potential scenario should, at a minimum, be considered by the authors.

6. PLOS authors have the option to publish the peer review history of their article (what does this mean?). If published, this will include your full peer review and any attached files.

Reviewer #1: No

Reviewer #2: **Yes: **Thiago Gomes Heck

Reviewer #3: No

---

## [Author Response · Author response to Decision Letter 0]

5 May 2021

Please find our responses to the reviewers' comments in the document named "Response to Reviewers."

---

## [Decision Letter · Decision Letter 1]

14 Jun 2021

Prevalence of Unmasked and Improperly Masked Behavior in Indoor Public Areas during the COVID-19 Pandemic: Analysis of a Stratified Random Sample from Louisville, Kentucky

PONE-D-21-05854R1

Dear Dr. Seyed M. Karimi,

We’re pleased to inform you that your manuscript has been judged scientifically suitable for publication and will be formally accepted for publication once it meets all outstanding technical requirements.

Kind regards,

Cosme F. Buzzachera, Ph.D.

Academic Editor

PLOS ONE

Reviewers' comments:

Reviewer's Responses to Questions

**Comments to the Author**

1. If the authors have adequately addressed your comments raised in a previous round of review and you feel that this manuscript is now acceptable for publication, you may indicate that here to bypass the “Comments to the Author” section, enter your conflict of interest statement in the “Confidential to Editor” section, and submit your "Accept" recommendation.

Reviewer #1: All comments have been addressed

Reviewer #2: All comments have been addressed

2. Is the manuscript technically sound, and do the data support the conclusions?

Reviewer #1: Yes

Reviewer #2: Yes

3. Has the statistical analysis been performed appropriately and rigorously? 

Reviewer #1: Yes

Reviewer #2: Yes

4. Have the authors made all data underlying the findings in their manuscript fully available?

Reviewer #1: Yes

Reviewer #2: Yes

5. Is the manuscript presented in an intelligible fashion and written in standard English?

Reviewer #1: Yes

Reviewer #2: Yes

6. Review Comments to the Author

Reviewer #1: The authors have properly address all my questions.

I have no further concerns.

The study results are certainly important in the context we are currently living.

Reviewer #2: The new version of the manuscript answered all the questions asked by all reviewers. The conclusion is straightforward and supported adequately by the methods, results, and discussion. I want to congratulate the authors for the relevant and exciting study that can be applied in different locations worldwide.

7. PLOS authors have the option to publish the peer review history of their article (what does this mean?). If published, this will include your full peer review and any attached files.

Reviewer #1: No

Reviewer #2: **Yes: **Thiago Gomes Heck

---

## [Editor Report · Acceptance letter]

12 Jul 2021

PONE-D-21-05854R1 

Prevalence of Unmasked and Improperly Masked Behavior in Indoor Public Areas during the COVID-19 Pandemic: Analysis of a Stratified Random Sample from Louisville, Kentucky 

Dear Dr. Karimi:

I'm pleased to inform you that your manuscript has been deemed suitable for publication in PLOS ONE. Congratulations! Your manuscript is now with our production department. 

Kind regards, 

on behalf of

Dr. Cosme F. Buzzachera 

Academic Editor

PLOS ONE